# Evaluating the Antifungal Potential of Botanical Compounds to Control *Botryotinia fuckeliana* and *Rhizoctonia solani*

**DOI:** 10.3390/molecules26092472

**Published:** 2021-04-23

**Authors:** Francisca Sempere-Ferre, Jordi Asamar, Vicente Castell, Josefa Roselló, M. Pilar Santamarina

**Affiliations:** 1Departamento de Estadística e Investigación Operativa Aplicadas y Calidad, Universitat Politècnica de València, Camino de Vera s/n, 46022 Valencia, Spain; 2Departamento de Ecosistemas Agroforestales, Universitat Politècnica de València, Camino de Vera s/n, 46022 Valencia, Spain; jasamarca@gmail.com (J.A.); jrosello@upvnet.upv.es (J.R.); mpsantam@eaf.upv.es (M.P.S.); 3Departamento de Producción Vegetal, Universitat Politècnica de València, Camino de Vera s/n, 46022 Valencia, Spain; vcastell@prv.upv.es

**Keywords:** antifungal activity, mixtures, thymol, carvacrol, cinnamaldehyde, botanical compound, *Botryotinia fuckeliana*, *Rhizoctonia solani*

## Abstract

The European Union is promoting regulatory changes to ban fungicides because of the impact their use has on the ecosystem and the adverse effects they can pose for humans. An ecofriendly alternative to these chemicals to fight against fungal species with low toxicity is essential oils and their compounds extracted from aromatic plants. The purpose of this study was to evaluate the in vitro antifungal capacity of the botanical compounds eugenol, carvacrol, thymol, and cinnamaldehyde, and the synergy or antagonism of their mixtures, against *Botryotinia fuckeliana* and *Rhizoctonia solani*. Different bioassays were performed at doses of 300, 200, 150, and 100 µg/mL using pure commercial compounds and their combination in potato dextrose agar culture medium. Growth rate and the mycelium growth inhibition parameters were calculated. Phenolic compounds and their combination inhibited the development of species at the different concentrations, with fungicidal or fungistatic activity shown under almost all the tested conditions. When comparing the growth rates of the species in the control plates and treatments, the statistical analysis showed that there were statistically significant differences. The mixture of compounds improved fungicidal activity against the studied species and at a lower concentration of monoterpenes.

## 1. Introduction

The European Union is promoting regulatory changes to ban many plant protection products, including fungicides [1]. This fact, together with its widespread use, has resulted in fungi resistance, causing the resurgence of new pathogens, generating toxicity to non-target organisms, impacting ecosystems, and posing adverse effects for humans. For all these reasons, alternative ways to combat these microorganisms must be sought [2].

Chemicals produced by various aromatic plants as a self-protection mechanism are a potential source of low-toxicity and environmentally friendly antimicrobial compounds. Of these phytochemicals, essential oils (EOs) and their components have aroused the scientific community’s interest in recent decades for their phytosanitary activity and their use in food preservation for their activity against a wide range of fungi, bacteria, viruses, and other organisms [3].

Essential oils are oily liquids obtained from different plant parts, such as flowers, buds, seeds, leaves, branches, bark, wood, fruit, and roots. They are complex mixtures of esters, aldehydes, ketones, terpenes, and other low-molecular-weight volatile compounds [4].

As fungicides, their use seems promising, but their performance is very fast and not persistent, owing to their easy volatilization. Moreover, they are easily degraded by oxidation, heating, or exposure to light, which poses a problem when applying them [5]. To overcome this problem, different alternative formulations are being developed that stabilize the product to increase the duration of its effect, reduce its volatilization, simplify its handling, and slow down its degradation in the environment, such as oil-in-water nano/microemulsions or polymeric micelles [6,7,8,9,10,11].

In addition, the antifungal activity of EOs depends on their chemical composition, which is determined by different factors like plant species and the part it was extracted from, the method to obtain it, geographical characteristics, and growing conditions. Hence the need to research its bioactive components of isolated forms and to test different mixtures to obtain formulations that replace phytosanitary products [12]. Of these botanical compounds, different monoterpenes stand out, such as carvacrol, eugenol, thymol, and cinnamaldehyde.

Eugenol is a pale-yellow oily liquid extracted from certain essential oils, especially from cloves (*Syzygium aromaticum*) and other vegetal species, such as nutmeg (*Myristica* sp.), cinnamon (*Cinnamomum verum*), and, in smaller quantities, from species like Jamaican pepper (*Pimenta* sp.), Indian pepper (*Piper* sp.), basil (*Ocimum* sp.), carrot seeds (*Daucus* sp.), or laurel (*Laurus* sp.) (Khalil et al., 2017). This chemical compound is approved as an active substance to be used as a pesticide (biocidal) by the European Commission’s Directorate-General of Health and Food Safety (Reg. (EU) No. 546/2013) [13].

Carvacrol is a monoterpenoid phenol with marked antifungal power that has been isolated in different proportions from EOs, such as 40–50% in oregano (*Origanum vulgare*), 50–80% in marjoram (*Origanum majorana*), 5–75% thyme (*Thymus* sp.), and 1–45% savory (*Satureja* sp.) [8,14,15,16].

Thymol belongs to the terpene group and is a carvacrol isomer. It is found in EOs of numerous species in the *Lamiaceae* family, especially in species of the genus *Thymus* at more than 50% of its oil composition, as in *T. zygis*, *T. piperella,* and *T. semperfilum* [17]. It is also abundant in oregano (*Origanum vulgare*) and marjoram (*O. majorana*) [18]. It is an active substance whose use as a pesticide (biocidal) is approved by the European Commission’s Directorate-General of Health and Food Safety (Reg. (EU) No. 568/2013) [19].

Cinnamaldehyde is a viscous organic compound in a liquid state of a pale-yellowish color capable of providing the characteristic taste and smell of cinnamon. It is found in the bark of the cinnamon tree and other species of the genus *Cinnamomum*. Cinnamon EO oil can contain up to 90% cinnamaldehyde [20]. It is a phenolic compound of some species, including cinnamon, and generally admitted as safe for use in food and employed in many foods as a flavoring [21].

The activity of these phenolic compounds has been investigated against different fungal species. Fungi are the most important group of phytopathogenic agents given the number and diversity of diseases that they cause [22]. Theoretically, there is no economically viable crop that is not affected by one microorganism or more, and these are specialized as being able to colonize a single plant species or species of a single genus. The wide parasitic versatility characterized by the diversity of susceptible plant species and plant organs that can be affected, the range of their symptomatology, and the diseases and damage that can be caused are quite remarkable. Therefore, the search for effective strategies against fungal problems has been a constant challenge [23].

*Botryotinia fuckeliana* (de Bary) Whetzel (anamorph *Botrytis cinerea* Pers) is a plant pathogen and a causative agent of the disease known as “gray mold”. The genus *Botrytis* is a group of widely distributed fungi, hence its substantial importance. This genus has more than 20 species, including *B. tulipae*, which affects tulip *B. squamosa*, onion and *B. fabae* bean. *B. fuckeliana* is one of the most relevant phytopathogenic fungi because of its infectious nature, which enables it to complete its infectious cycle on plants and their saprophytic nature which, in turn, allows it to live on different plant material: senescent, dead, or previously infected by other pathogens [24].

The soil-borne fungus *Rhizoctonia solani* JG Kühn is a pathogen of many plants and severely damages crops worldwide. In the potato plant, it is able to develop at very different temperatures in warm, temperate, and cold areas, and causes considerable damage to tuber shoots [25]. Symptomatology is characterized by lesions appearing on roots, stolons, and necks of plants, and also by the appearance of isolated or mass sclerotics known as black scurf in tubers [26]. Strains of *R. solani* from the anastomosis group (AG) 3 attack potatoes, which results in severe yield losses and downgrades in production [27].

The aims of this work were to (a) study the in vitro antifungal activity of botanical monoterpenes eugenol, carvacrol, thymol, and cynamaldehyde against *Botryotinia fuckeliana* and *Rhizoctonia solani*, and (b) investigate the synergy or antagonism of their mixtures against fungal species.

## 2. Results

### 2.1. Monoterpenes Antifungal Activity

Eugenol applied at a dose of 300 µg/mL to culture medium brought about a reduction in the growth rate of both *Botryotinia fuckeliana* and *Rhizoctonia solani*. Significant differences were found when comparing growth rates to the control group (Table 1 and Table 2) (Figure 1a and Figure 2a).

The mycelium growth inhibition (MGI) of *Botryotinia fuckeliana* and *R. solani* was 33.96 and 48.78%, respectively. This bioactive component was not tested at lower doses, but research into its synergy with other EO compounds was not rejected.

Carvacrol, cinnamaldehyde, and thymol monoterpenes were tested at doses of 300, 200, 150, and 100 µg/mL.

Carvacrol completely inhibited the fungal growth of *B. fuckeliana*, except at 100 µg/mL, at which it began to grow from day 4 when the species’ growth rate lowered to 1.48 mm·day ^−1^ (Figure 1b). MGI was 100% at all the assayed doses, except at 100 µg/mL (87.97%). When fungal growth inhibition was 100%, the inoculation fungus disc was transferred from media to a control plate (with no active compound) (Table 1) (Figure 1a).

The study of the fungicidal or fungistatic activity of the different tested carvacrol concentrations in culture medium showed fungicidal activity under all the conditions except at 100 and 150 µg/mL, when the fungus was not eradicated and fungistatic activity was recorded. Indeed, the minimum fungicidal dose in vitro of carvacrol against *B. fuckeliana* was 200 µg/mL.

The MGI was 100% when the effect of the different doses of carvacrol were tested against *R. solani* showing fungicidal activity (Table 2) (Figure 2b) (Table 3).

After adding cinnamaldehyde to culture medium, the MGI of *B. fuckeliana* was 37.48 and 48.68% for the 100 µg/mL dose and the 150 µg/mL dose, respectively, with an MGI of 100% for doses of 200 and 300 µg/mL. The minimum monoterpene concentration with fungicidal activity against this species was the same as that obtained with carvacrol (Table 3).

When the effect of this phenylpropanoid against *R. solani* was assayed, all the doses showed fungistatic activity, and the species’ growth rate lowered. Fungus growth decreased as the active ingredient concentration in culture medium did, with 90.29, 66.79, 46.60, and 31.55% for the 300, 200, 150 and 100 µg/mL doses, respectively (Table 2) (Figure 2c).

A fungicidal effect at all the tested concentrations was observed when the botanical compound thymol was added to medium and tested against *Rhizoctonia solani*, with zero growth rates and an MGI of 100%. When the phenolic compound was tested against *B. fuckeliana,* fungal development was completely inhibited at the higher doses of 300 and 200 µg/mL, but not at the lower doses of 150 and 100 µg/mL. At these last concentrations, thymol had a fungistatic effect, and the species’ growth rate slowed down compared to the control, with an MGI of 87.09 and 57.52%, respectively (Table 1 and Table 2) (Figure 1d and Figure 2d) (Table 3).

When comparing the growth rates of *B. fuckeliana* and *R. solani* in both the control plates and the different culture media with the concentrations of the botanical compounds, the statistical analysis showed significant differences under all the conditions (*p* < 0.05).

### 2.2. Synergistic Effect of the Botanical Active Compounds

Mixtures of the botanical phenolic compounds against *B. fuckeliana* and *R. solani* gave very satisfactory results at the 150 µg/mL concentration.

Eugenol, which was initially discarded at doses below 300 µg/mL, was included in this study to investigate whether its application at 150 µg/mL in synergy with other active ingredients would improve its results.

All the tested botanical active mixtures completely inhibited the fungal growth of *B. fuckeliana*, with an MGI of 100%, and displayed a fungicide effect, except for the cinnamaldehyde + eugenol mixture, for which slow growth was recorded (Table 4) (Figure 3).

Compared to *R. solani,* the tested mixtures completely inhibited fungal growth with an MGI of 100% and had a fungicide effect, except for the cinnamaldehyde + eugenol mixture, in which the fungus began to grow on day 3 at a growth rate of 3.3 mm·day^−1^ and with an MGI of 82.47%.

## 3. Discussion

This manuscript investigated the antimicrobial capacity of bioactive components eugenol, carvacrol, cinnamaldehyde and thymol, and their mixtures, against the fungal species *Botryotinia fuckeliana* and *Rhizoctonia solani.*

Phenolic compounds and their combination inhibited the development of the described species at different concentrations and displayed fungicidal or fungistatic activity under almost all the tested conditions.

The minimum concentration at which carvacrol and thymol fungicidal activity was recorded was 200 μg/mL for *Botryotinia fuckeliana* and 100 mg/mL for *Rhizoctonia solani*. In a study carried out by Ochoa et al. to investigate activity within a concentration range of 100, 150, 200, 400, 800, and 1600 μg/mL, no development of *F. sambucinum* was recorded at 150 μg/mL for carvacrol and at 400 μg/mL for thymol [28].

Thymol, carvacrol, eugenol, and cinnamaldehyde have been successfully tested against several species with different MICs and minimum fungicidal concentrations (MFC) against food-relevant fungi like *Aspergillus niger*, *A. fumigatus*, *A. flavus*, *A. ochraceus*, *Alternaria alternata*, *Cladosporium* sp., *Penicillium citrinum*, *P. chrysogenum*, *F. oxysporum*, *Molinia* sp., and *Botryotinia fuckeliana* [28,29,30,31]. For the last plant pathogen, the MIC and MFC values were 300 and 350 μg/mL in thymol, 300 and 350 μg/mL for carvacrol, and 450 and 500 μg/mL for eugenol [29]. In another study, the application of a carvacrol vapor atmosphere to *Botryotinia fuckeliana* inoculated in potato dextrose agar (PDA) inhibited fungal development under all the tested conditions, and this compound has been shown to inhibit the germination of its conidia [32].

When 20 monoterpenes were tested against *Rhizoctonia solani*, thymol was the compound that displayed the most activity, and was comparable to a reference fungicide like carbendazim [33].

Different studies have determined that thymol has greater fungicidal activity than carvacrol [34,35]. When 41 monoterpenes were tested individually or in combination against *Rizophus stolonifer* and *Absidia coerulea*, eugenol, and carvacrol were the compounds with the most marked activity, and carvacrol obtained better results at the doses of 44.94 and 50.83 μg/mL [36].

In the present research work, thymol and carvacrol were equally active against *Rhizoctonia solani*. When considering the different assayed doses, carvacrol was better than thymol when tested against *Botryotinia fuckeliana*. The discrepancy in the obtained results compared to other studies could be due to different factors like testing conditions and the experimentation methodology, assayed strains, the doses of the employed active principles, etc.

Different mechanisms of action of active principles and EOs on fungal species have been described, such as ruptured cell wall and membrane disruption, inhibition of chitin synthesis, ROS accumulation, mitochondrial dysfunction, and inhibition of some specific enzyme activities [31,33,37]. Carvacrol has been found to produce morphological changes *in Rhizopus stolonifer*, where hyphae were severely damaged, the cytoplasm structure disappeared and residual cell contents flocculated sideways [38]. The same compound on *Colletrichum* species brought about inhibition in spore germination, loss of cell wall integrity and function, decreased cell metabolism, and ROS accumulation [12]. Cinnamaldehyde resulted in alteration to not only permeability, but also to the structure of *Geotrichum citri-aurantii* [39]. When monoterpene was tested on *Penicillium expansum*, the compound brought about alterations like cell folding, loss of cell wall integrity, disrupted plasma membrane, mitochondria destruction, and the absence of intracellular organelles [36,37]. In this sense, it seems that the mechanisms involved in the biotoxicity of the compounds may vary in different conditions [40].

The Fungicide Resistance Action Committee (FRAC) classified carvacrol and thymol according to their mode of action in the F7 group (transport or synthesis of lipids/function or membrane integrity). This group includes those active substances that alter the synthesis of lipids and the integrity of cell membranes by acting on their permeability and affecting glycolypid formation. They act as cell membrane disruptors (FRAC code: 46).

The mixture of compounds improved fungicidal activity against the studied species when active principles were tested individually, which requires a lower concentration. These results coincide with former studies carried out by other researchers [28,36]. The mechanisms of this synergism are not yet clear [41]

Eugenol, cinnamaldehyde, carvacrol, and thymol are natural bioactive compounds extracted from oils like clove, cinnamon, satureja, thyme, laurel, and oregano. These monoterpenes have shown their effectiveness in fungi such as: *Fusarium* sp., *Alternaria* sp., *Penicillium* sp., *Curvularia* sp., *Sclerotinia* sp. *Phytoptora* sp. *Rhizoctonia* sp., and *Aspergillus* sp. [17,42,43,44]. When the EOs of *Thymbra spicata*, *Satureja thymbra*, *Salvia fruticosa*, *Laurus nobilis*, *Mentha pulegium*, *Inula viscosa,* and *Pimpinella anisum* were analyzed, antifungal activity was found to be due to the phenolic fraction of the oils composed of carvacrol and thymol [35]. These phenolic compounds have also shown activity against the mycotoxins produced by fungi [3,45].

The EOs of *Carum carvi*, *Cinnamomum zeylanicum*, *Citrus aurantifolia*, *Citrus latifolia*, *Citrus limon*, *Citrus × limonia*, *Melaleuca alternifolia*, *Origanum majorana*, *Poliomintha longiflora*, *Rosmarinus officinalis*, *Satureja khusiztanica*, *Satureja montana*, and *Mentha longifolia* have been effective in controlling *Botryotinia fuckeliana*, while the oils of *Mentha piperita*, *Satureja montana*, and *Mentha longifolia* have proven effective been against *Rhizoctonia solani* [3,8].

The botanical active principles and the EOs are presented as an alternative to chemical compounds for the control of phytopathogenic and postharvest fungi and food preservation. Carvacrol and thymol have been evaluated as being safe for humans and recognized as food additives by the European Food Safety Agency [46]. It is necessary to look for formulas that respect the environment and are harmless at low doses to reduce the economic losses generated by the development of these microorganisms.

## 4. Materials and Methods

### 4.1. Fungi

The species *Botryotinia fuckeliana* CECT 2100 and *Rhizoctonia solani* CECT 2819 were supplied by the CECT (Spanish Type Culture Collection). Samples of these fungi were maintained at the Botany Laboratory of the Department of Agroforestry Ecosystems.

### 4.2. Botanical Compounds

Pure chemical compounds with different purities isolated from EOs were supplied by Aldrich (ref): thymol SLBD1228V (≥98.5%), carvacrol MKBN3724V (99%), cinnamaldehyde MKBX1392V (95%), and eugenol MKBR1493V (98%).

### 4.3. Antifungal Activity

Both the individual pure botanical compounds and mixtures were dissolved, mixed, and homogenized by agitation in flasks with culture medium PDA/Tween 20 (0.1%), previously sterilized. While still liquid, active principles were added at different concentrations. Initially a dose of 300 μg/mL was assayed. Given the results from the first trial, eugenol was rejected for subsequent bioassays, and cinnamaldehyde, carvacrol, and thymol were studied at 200, 150, and 100 μg/mL. The mixture of monoterpenes was prepared at a concentration of 150 μg/mL.

Media were distributed in 90 × 15 and 150 × 15 mm Petri plates. Fungi explants (8 mm diameter) from a 7-day culture were inoculated in the center of each plate and were incubated at 25 °C for 6 days.

Petri plates were examined daily, and the diameter of the growing colonies was measured in two directions at right angles to each other. Measurements were taken for 6 days. In all cases, the linear regression of increased radius (mm) against time (in days) was used to obtain the growth rates (mm·day^−1^) for each set of treatment conditions.

For each condition (botanical active compound, dose, and fungal species), five plates were inoculated and the experiment was repeated three times.

### 4.4. Mycelial Growth Inhibition (MGI)

Mycelial growth inhibition was calculated on day 6 by Equation (1) [47]:MGI = [(DC − DO)/DC] × 100(1)
where CD is the average diameter of colonies in non-treated plates (without carvacrol, eugenol, or cinnamaldehyde) and OD is the average diameter of colonies with botanical compounds alone or in their mixtures.

### 4.5. Fungicide or Fungistatic Activity

After determining the antifungal activity of the active principles and their mixtures, for those whose results showed 100% inhibition, their fungicide and/or fungistatic effect was checked. Of those plates that gave 100% inhibited fungal growth after the 7-day incubation, the same fungal inoculation disc (8 mm) was taken and sown in the center of the Petri plates containing only PDA. The experiment was incubated at 25 °C for 7 days. The growth diameter of each fungus was measured after the incubation period.

### 4.6. Statistical Analysis

The analysis of variance (ANOVA) was used to determine the influence of carvacrol, eugenol, and cinnamaldehyde, and their mixtures, on the mycelial growth of *R. solani* and *B. fuckeliana*. Fisher’s LSD and Tukey’s HSD intervals were used to compare the effect of simple factors, species, and treatment, and their interaction, with significance values of *p* < 0.05. The Statgraphics XVIII software (Stat Point, Inc., Herndon, VA, USA) was used in the study.

## Figures and Tables

**Figure 1 molecules-26-02472-f001:**
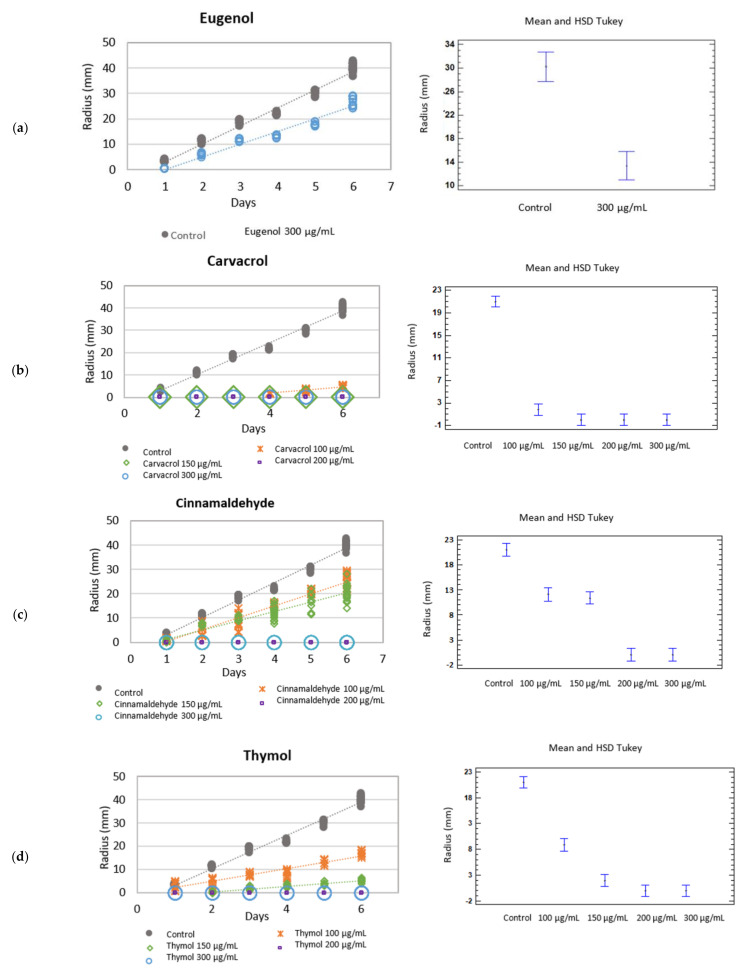
Left: Growth rate (mm·day^−1^) of *Botryotinia fuckeliana* in the different botanical actives compounds. Right: Mean and Tukey’s HSD, *n* = 30. (**a**) eugenol, (**b**) carvacrol, (**c**) cinnamaldehyde, (**d**) thymol.

**Figure 2 molecules-26-02472-f002:**
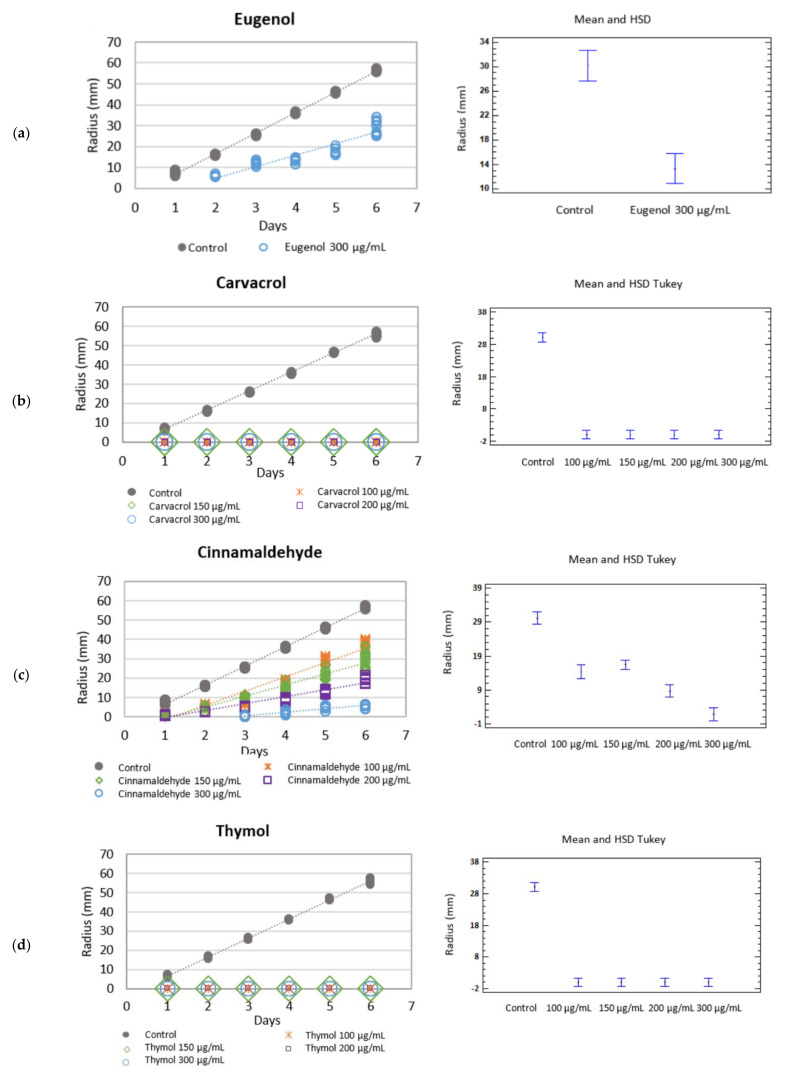
Left: Growth rate (mm·day^−1^) of *Rhizoctonia solani* in the different botanical active compounds. Right: Mean and Tukey’s HSD, *n* = 30. (**a**) eugenol, (**b**) carvacrol, (**c**) cinnamaldehyde, (**d**) thymol.

**Figure 3 molecules-26-02472-f003:**
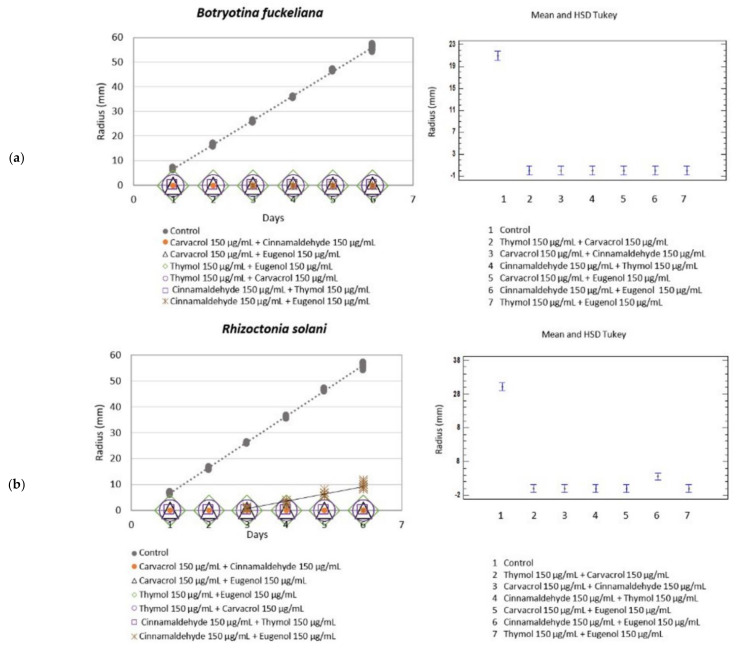
Left: Growth rate (mm·day^−1^) of *B. fuckeliana* (**a**) and *R. solani* (**b**) in the different botanical actives mixtures. Right: Mean and Tukey’s HSD, *n* = 30.

**Table 1 molecules-26-02472-t001:** Effect of pure botanical active compounds and their mixtures on the growth rate of *Botryotinia fuckeliana* and mycelial growth inhibition (MGI).

Botanical Active Compounds	Dose µL/mL	GR (R^2^)	d.s.	Mean Ø Day 6 (mm)	MGI (%)
Control	-	7.11 (0.985)	a	39.90	-
Eugenol	300	5.00 (0.969)	b	26.35	33.96
Carvacrol	300	0	c	0	100
	200	0	c	0	100
	150	0	c	0	100
	100	1.48 (0.866)	b	4.80	87.97
Cinnamaldehyde	300	0	c	0	100
	200	0	c	0	100
	150	3.79 (0.882)	b	20.48	48.68
	100	4.89 (0.949)	b	24.94	37.48
Thymol	300	0	c	0	100
	200	0	c	0	100
	150	1.22 (0.872)	c	5.15	87.09
	100	2.72 (0.870)	b	16.95	57.52
**Mixtures**					
Eug + Thy	150 + 150	0	b	0	100
Thy + Car	150 + 150	0	b	0	100
Car + Cin	150 + 150	0	b	0	100
Cin + Eug	150 + 150	0	b	0	100
Car + Eug	150 + 150	0	b	0	100
Thy + Cin	150 + 150	0	b	0	100

GR: growth rate (R^2^); d.s.: homogeneous groups due to significant differences; Eug: eugenol; Thy: thymol; Car: carvacrol; Cin: cinnamaldehyde.

**Table 2 molecules-26-02472-t002:** Effect of pure botanical active compounds and their mixtures on the growth rate of *Rhizoctonia solani* and mycelial growth inhibition (MGI).

Botanical Active Compounds	Dose µL/mL	GR (R^2^)	d.s.	Mean Ø Day 6 (mm)	MGI (%)
Control	-	9.89 (0.998)	a	56.61	
Eugenol	300	5.54 (0.872)	b	28.99	48.78
Carvacrol	300	0	b	0	100
	200	0	b	0	100
	150	0	b	0	100
	100	0	b	0	100
Cinnamaldehyde	300	1.80 (0.833)	d	5.50	90.29
	200	3.60 (0.955)	c	18.80	66.79
	150	5.73 (0.918)	b	30.23	46.60
	100	7.40 (0.945)	b	38.75	31.55
Thymol	300	0	b	0	100
	200	0	b	0	100
	150	0	b	0	100
	100	0	b	0	100
**Mixtures**					
Eug + Thy	150 + 150	0	c	0	100
Thy + Car	150 + 150	0	c	0	100
Car + Cin	150 + 150	0	c	0	100
Cin + Eug	150 + 150	3.31 (0.957)	b	9.93	82.47
Car + Eug	150 + 150	0	c	0	100
Thy + Cin	150 + 150	0	c	0	100

GR: growth rate (R^2^); d.s.: homogeneous groups due to significant differences; Eug: eugenol; Thy: thymol; Car: carvacrol; Cin: cinnamaldehyde.

**Table 3 molecules-26-02472-t003:** Fungicidal or fungistatic activity of the pure botanical active compounds.

Botanical Active Compounds	*B. fuckeliana*				*R. solani*			
	Dose (µg/mL)				Dose (µg/mL)			
	100 µg/mL	150 µg/mL	200 µg/mL	300 µg/mL	100 µg/mL	150 µg/mL	200 µg/mL	300 µg/mL
**Eugenol**	-	-	-	fungistatic	-	-	-	fungistatic
**Carvacrol**	fungistatic	fungistatic	fungicide	fungicide	fungicide	fungicide	fungicide	fungicide
**Cinnamaldehyde**	fungistatic	fungistatic	fungicide	fungicide	fungistatic	fungistatic	fungistatic	fungistatic
**Thymol**	fungistatic	fungistatic	fungicide	fungicide	fungicide	fungicide	fungicide	fungicide

**Table 4 molecules-26-02472-t004:** Fungicidal or fungistatic activity of botanical active mixtures.

Botanical Active Mixtures	*B. fuckeliana*	*R. solani*
Thymol 150 µg/mL + Carvacrol 150 µg/mL	fungicide	fungicide
Carvacrol 150 µg/mL + Cinnamaldehyde 150 µg/mL	fungicide	fungicide
Carvacrol 150 µg/mL + Eugenol 150 µg/mL	fungicide	fungicide
Cinnamaldehyde 150 µg/mL + Thymol 150 µg/mL	fungicide	fungicide
Cinnamaldehyde 150 µg/mL + Eugenol 150 µg/mL	fungistatic	slow growth
Thymol 150 µg/mL + Eugenol 150 µg/mL	fungicide	fungicide

## Data Availability

Data sharing is not applicable to this article.

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
