# Peer review of "Evaluating the Antifungal Potential of Botanical Compounds to Control Botryotinia fuckeliana and Rhizoctonia solani"

_molecules, 2021, doi:10.3390/molecules26092472_

Round 1
Reviewer 1 Report
Overall, the study presented is simple, but effectively presents the effects of bioadditives/essential oils to control fungal contamination. Overall, the study is well written, easy to follow and effective. I recommend to be published with minor corrections (see below).
Specific comments:
Figure 1. if there was extensive statistics applied to this figure, please include error bars and indicate the 'n' number in the caption. Same for other figures.
Table 2. Row 1. The value 0.998 is erroneously typed as 0,998
Author Response
Gracias por las contribuciones realizadas en el artículo. De acuerdo con sus indicaciones, se han realizado modificaciones.
El texto modificado aparece en rojo, mientras que el texto que se va a eliminar está subrayado en amarillo y tachado.
Se han modificado algunos aspectos del lenguaje. Se han modificado algunos aspectos del lenguaje. En este sentido, se adjunta certificado de la revisión realizada por profesionales nativos.
Además, las barras de error y el tamaño de la muestra n se han incluido en las diferentes figuras.
Tabla 2. Fila. Se ha cambiado el valor 0,998.

Reviewer 2 Report
The manuscript “Evaluating the antifungal potential of botanical compounds to control Botryotinia fuckeliana and Rhizoctonia solani” analyse the in vitro antifungal activity of some botanical compounds, and the synergy or antagonism of their mixtures, against two phytopathogenic fungi. The authors demonstrated that phenolic compounds and their combination inhibited the development of Botryotinia fuckeliana and Rhizoctonia solani, with fungicidal or fungistatic activity shown under almost all the tested conditions.
It is well structured and the methods used are correct.
I suggest that the paper could be considered suitable for publication after minor English revisions.
Author Response
Gracias por las contribuciones realizadas en el artículo.
Hay algunas modificaciones de acuerdo con los comentarios realizados por el revisor 1 y 3.

Reviewer 3 Report
The work presents a very nice work of the use of essential oils agains pathogens for vegetal species. The results are sound and can have impact to the community. However, authors should address some major concerning before publication:
-authors should differentiate between essential oils and essential oils compounds. Authors use both terms as synonyms
-the description of the different essential oil compounds is not proper for a research article. Authors should remove it from the text
-why was Eugenol tested to only one dose? A proper explanation should be included.
-about synergism authors should include: ACS Sustainable Chemistry and Engineering 2020, 8, 10995–11006 and Bulletin of Insectology 2020, 73, 153-160
-why do the authors use directly the essential oil compounds? Have any idea about their potential transported in any type of platform?
-the work by Lucia and Guzmán should be included in the reference list Advances in Colloid and Interface Science 2021, 287, 102330
-about the effect of different essential oils, the referenc Peer J 2017, 5, e3171 should be included.
-the discussion appears elusive, more details about the origin of the effects should be included
Author Response
Thanks for the contributions made in the article. In accordance with your indications, modifications have been made.
The modified text appears in red while the text to be deleted has been underlined in yellow and struck through.
Some aspects of the language have been modified. Some aspects of the language have been modified. In this sense, a certificate of the review carried out by native professionals is attached.
Table 2. Row. The value 0.998 has been changed
The introduction has been modified and the references suggested by the reviewer have now been included.
the work by Lucia and Guzmán should be included in the reference list Advances in Colloid and Interface Science 2021, 287, 102330 Included.Line 57
-about the effect of different essential oils, the referenc Peer J 2017, 5, e3171 should be included. Included.Line 57.
The description of the different essential oil compounds is not proper for a research article. Authors should remove it from the text. Part of the paragraph has been removed. It is marked in yellow and crossed out.
why do the authors use directly the essential oil compounds? Have any idea about their potential transported in any type of platform? References to biofilms and compound platforms have been increased.
The results have been improved by modifying the figures and including the error bars.
Authors should differentiate between essential oils and essential oils compounds. Authors use both terms as synonyms. Some modifications have been made in the discussion so that the terms essential oils and botanical compounds are clearly differentiated. Line 320-342.
-why was Eugenol tested to only one dose? A proper explanation should be included.
The MGI of eugenol against B. fuckeliana and R. solani was 48.78% and 39.90%, therefore it makes no sense to lower the dose due to its high percentage of inhibition, but its study in mixtures was not ruled out.
This explanation has been included in the Material and Methods section. 4.3 Antifungal activity.
Initially a dose of 300 μg/mL was assayed. Given the results from the first trial, eugenol was rejected for subsequent bioassays, and cinnamaldehyde, carvacrol and thymol were studied at 200, 150 and 100 μg/mL. The mixture of monoterpenes was prepared at a concentration of 150 μg/mL.
And Line 130
2.1. Monoterpenes antifungal activity
Eugenol applied at the dose of 300 µg/ mL to culture medium brought about a re-duction in the growth rate of both Botryotinia fuckeliana and Rhizoctonia solani. Signifi-cant differences were fund when comparing growth rates to the control group (Tables 1 & 2) (Figures 1a &2a).
The mycelium growth inhibition (MGI) of Botryotinia fuckeliana and R. solani was 33.96% and 48.78%, respectively. This bioactive component was not tested at lower doses, but research into its synergy with other EO compounds was not rejected.
Discusion:
-the discussion appears elusive, more details about the origin of the effects should be included. Modified the paragraph.
-about synergism authors should include: ACS Sustainable Chemistry and Engineering 2020, 8, 10995–11006 and Bulletin of Insectology 2020, 73, 153-160. Included. Line 317.

Round 2
Reviewer 3 Report
Authors have improved the manuscript, and now It is fully publishable